# Is the Use of Monensin Another Trojan Horse for the Spread of Antimicrobial Resistance?

**DOI:** 10.3390/antibiotics13020129

**Published:** 2024-01-28

**Authors:** Cristina Carresi, Romano Marabelli, Paola Roncada, Domenico Britti

**Affiliations:** 1Veterinary Pharmacology Laboratory, Department of Health Sciences, Interregional Research Center for Food Safety and Health IRC-FSH, University “Magna Graecia” of Catanzaro, 88100 Catanzaro, Italy; 2World Organization for Animal Health, 75017 Paris, France; r.marabelli@woah.org; 3Department of Health Sciences, University “Magna Graecia” of Catanzaro, 88100 Catanzaro, Italy; roncada@unicz.it (P.R.); britti@unicz.it (D.B.); 4Interdepartmental Center Veterinary Service for Human and Animal Health, University “Magna Graecia” of Catanzaro, CISVetSUA, 88100 Catanzaro, Italy

**Keywords:** ionophores, antimicrobial resistance, environment, One Health approach

## Abstract

Antimicrobial resistance (AMR) is a complex and somewhat unpredictable phenomenon. Historically, the utilization of avoparcin in intensive farming during the latter part of the previous century led to the development of resistance to vancomycin, a crucial antibiotic in human medicine with life-saving properties. Currently, in the European Union, there is a growing reliance on the ionophore antibiotic monensin (MON), which acts both as a coccidiostat in poultry farming and as a preventative measure against ketosis in lactating cows. Although many researchers claim that MON does not induce cross-resistance to antibiotics of clinical relevance in human medicine, some conflicting reports exist. The numerous applications of MON in livestock farming and the consequent dissemination of the compound and its metabolites in the environment require further investigation to definitively ascertain whether MON represents a potential vector for the propagation of AMR. It is imperative to emphasize that antibiotics cannot substitute sound animal husbandry practices or tailored dietary regimens in line with the different production cycles of livestock. Consequently, a rigorous evaluation is indispensable to assess whether the economic benefits associated with MON usage justify its employment, also considering its local and global environmental ramifications and the potential risk of instigating AMR with increased costs for its control.

## 1. Introduction

Since the inception of the antimicrobial era, it has been evident that antimicrobial resistance (AMR) is an anticipated phenomenon requiring careful management.

Bacterial resilience in the face of antibiotic stress could be defined by two key factors: resistance and tolerance. The use of various antibiotics has been demonstrated to increase the mutation rates within bacterial genomes [1,2]. Consequently, the survival of bacterial populations through tolerance can serve as an environment conducive to the subsequent emergence of AMR and possible cross-resistance or co-selection to two or more drugs. Regrettably, in economic processes, when the pharmaceutical and chemical industries identify a promising molecule, they seek to expand its application across a wider range of species. Within the same species, they aim to combat more diseases or employ it in various contexts involving humans, plants, and animals. If the molecule or any of its derivatives/metabolites become pervasive and resistant to complete degradation into inert forms, it may exert unforeseen effects on microorganisms seemingly outside the molecule’s intended scope, thereby leading to unanticipated consequences.

Towards the end of the previous century, it became apparent that the use of substances like avoparcin in intensive farming had the potential to foster resistance to vital drugs used in human medicine, such as vancomycin. Immediate and progressively stringent measures were taken to mitigate the resultant harm [3]. Although vancomycin and avoparcin share structural similarities as glycopeptides, making it foreseeable that cross-resistance could emerge among these drugs, it cannot be ruled out that cross-resistance may also develop towards chemically unrelated pharmaceuticals [1]. Presently, there is a growing utilization of monensin (MON), an antimicrobial belonging to the ionophore class. It is approved for inclusion in medicated feed for lactating dairy cattle in the United States [4]. In the European Union, MON was previously authorized as a feed additive to enhance growth and feed efficiency in cattle, except lactating cows (Council Regulation (EC) No. 1831/2003). The use of MON in cattle was phased out in January 2006. Monensin has been re-evaluated and authorized as a feed additive for the control of coccidiosis in poultry (Council Regulation (EC) No. 1356/2004; amended by Council Regulation (EC) No. 108/2007) [5,6,7]. In 2013, the European Medicines Agency granted authorization for the use of MON in dairy cows to prevent ketosis, particularly in transition cows [8]. The multifaceted applications of MON in livestock demand comprehensive consideration and contemplation within the medical and veterinary sectors. Consequently, this prompts the question: Can the use of MON serve as another Trojan horse for the spread of antimicrobial resistance?

### 1.1. Chemical Structure and Pharmacological Properties

Monensin, also called monensic acid (2-[5-Ethyltetrahydro-5-[tetrahydro-3-methyl-5-[tetrahydro-6-hydroxy-6-(hydroxymethyl)-3,5-dimethyl-2H-pyran-2-yl]-2-furyl]-2-furyl]-9-hydroxy-β-methoxy-α,γ,2,8-tetramethyl-1,6-dioxaspiro[4.5]decane-7-butyric acid) is an ionophore antibiotic belonging to the polyether monocarboxylic acid chemical family, a subclass of polyketides. It was discovered in 1967 by Agtarap and co-workers as a product of fermentation of *Streptomyces cinnamonensis* bacteria [9] and was first synthetized in 1979 by Kishi [10]. Several homologues were subsequently isolated but monensin A (hereafter MON) remains the most widely used. The ionophore molecule contains six oxygen atoms, five of which likely participate in the complexation of cations. It is maintained in a pseudo cyclic conformation due to the presence of bifurcated intramolecular hydrogen bonds formed between carboxyl group on one side of the molecule and two hydroxyl groups on the opposite side and the structure is stabilized by inter- and intramolecular hydrogen bonds (Figure 1) [11].

Monensin displays many biological and pharmacological properties of scientific interest.

It shows cellular growth inhibiting effects, both in plant and animal cells, by blocking the intracellular transport of proteins and other products of the Golgi apparatus [12]. In animal cells, MON induces mitochondrial damage and reduces the process of endocytosis [13]. It also damages the external structures of the cell surface, by reducing the secretion of proteoglycans, collagen, procollagen, and fibronectin [14]. Interesting studies highlighted the antiviral activity of MON, capable of inhibiting the penetration and replication of the virus by interfering with the synthesis of DNA [15] and an antimalarial action given by the establishment of an acid environment and a subtraction of nutrients hostile to the parasite growth [16]. Since its discovery, MON has been used worldwide in industrial poultry farming as a coccidiostat, due to its demonstrated antiprotozoal effects [17].

### 1.2. Antimicrobial Activity

Among all, the antimicrobial activity of MON is the most studied and has been attributed to the molecule’s ability to insert itself into the cell membrane and induce changes in the maintenance of pH and in the sodium–potassium balance within the cellular structure, which can lead to a reduction in the secretion and transport of substances important for the proper functioning of the cell [18,19,20].

In particular, MON as a cyclic polyether molecule forms complexes with metal cations, especially sodium ions (Na^+^) facilitating the transport of these ions across the lipid membranes of bacterial and protozoal cell membranes. Binding of MON to Na^+^ alters the normal Na^+^ concentration gradient across the cell membrane by affecting its electrochemical potential, which is essential for several cellular processes, including energy generation and nutrient transport. Indeed, MON can specifically interfere with a critical component of the ATP synthesis, the proton motive force (PMF), and other energy-dependent processes, decreasing cellular energy levels. Furthermore, its ability to interfere with Na^+^ transport, which plays a crucial role in the transport of nutrients, such as amino acids, into bacterial cells, leads to disruption of nutrient uptake and a reduction of essential nutrients available to the microorganism, ultimately leading to cell death [19].

When added to ruminant feed, MON, due to the changing in the sodium–potassium balance of the bacterial cells of the rumen microbiota, forces them to expend more energy to restore the balance leading to a reduction in growth or death [21].

In the rumen, this phenomenon mainly affects Gram-positive bacteria of the genera *Micrococcus*, *Bacillus*, and *Staphylococcus* in favor of Gram-negative, and causes a shift in rumen bacterial population [22]. This shift alters rumen fermentation to increase propionate production and reduce carbon dioxide and methane loss [23], thus improving the efficiency of energy metabolism with numerous benefits for production and animal health [21,24].

Stimulation of food metabolism in ruminants, associated with significant modifications of the intestinal microflora leads to the assimilation of increasing amounts of digested proteins. This ensures faster livestock growth and leads to more exploitation in terms of production [25,26]. Due to its metabolic, antibacterial and coccidiostatic properties, MON has been used for years as a non-hormonal growth promoting agent in veterinary practice.

Although the use of MON as a coccidiostat in poultry, a growth promoter or anti-ketotic agent in cattle is relatively safe at the recommended doses, its misuse or abuse to obtain better results in a shorter time can lead to serious consequences due to a narrow safety margin in some animal species [27]. Indeed, the ionophore activity of MON is responsible for secondary pharmacological effects in target and non-target species that mainly affect the cardiovascular system [28,29]. Monensin exerts cardiac inotropic activity through membrane current mechanisms and induces changes in subcellular organelles of cardiac and skeletal muscle. Sensitivity to MON side effects has been found to vary greatly between species [7].

Several accidental poisonings resulting from MON overdosage, misuse, and mixing errors in feed preparation, have been recorded in different animal species, such as cattle, sheep, pigs, and horses [30,31,32,33]. Signs of ascites, hydrothorax, hepatomegaly, and focal areas of myocardial and skeletal muscle lesions were also observed in water buffaloes fed with MON-containing feed [34].

The use of MON in dairy cows for the prevention of ketosis is interesting [8]. Ketosis is a metabolic disorder in which blood glucose levels are low and substances called ketones (such as acetoacetic acid and β-hydroxybutyrate) gather in the blood. In 2013, the Committee for Medicinal Products for Veterinary Medicine (CVMP) concluded that the benefits of MON-based veterinary medicine outweigh the risks for the approved indication, thus recommending the granting of marketing authorization. To date, the only pharmaceutical form approved in Europe is a continuous release intraruminal device (Kexxtone, Elanco GmbH, Cuxhaven, Germany) [35], whose use is quickly spreading.

Monensin (Kexxone) works by changing the microbial population in the rumen, resulting in an increase in propionate-producing bacteria and a reduction in β-hydroxybutyrate (BHB) in the blood.

β-hydroxybutyrate concentration of 10 mg/dL is the critical threshold to predict ketosis, displaced abomasum, and metritis in dairy cows [36]. These modifications improve liver function and energy production in the cow’s body, reducing the incidence of peripartum disease. The review of existing literature confirmed that MON delivered as a controlled-release capsule during the transition period has effects of different magnitude compared to other forms, doses, or durations of administration. Most studies agree on the anti-ketotic effects of this treatment, documenting the absence of negative impact on the production and composition of milk and cheese [37].

## 2. Peer-Reviewed Results

To date, the published peer-reviewed literature does not fully clarify the effects on the possible negative impact of such massive treatment on AMR, but some interesting scientific evidence has caused the great challenge of MON resistance to be further explored [1,35,38].

Therefore, from a One Health perspective relating to the risks of the use of MON in the development of AMR, it is important to provide an overview of the literature on the effects of this ionophore on the ruminal microflora and in the ecotoxicological field.

### 2.1. Antimicrobial Resistance

Resistance to MON has been studied by detecting sensitive and mutant colonies of some rumen bacterial species isolated from sheep, cattle, and buffaloes [37]. Through the Kirby–Bauer disk diffusion susceptibility test, the authors showed that the number of bacterial mutants of the different ruminant species, for all the antibiotics tested, was the highest in buffalo, followed by cattle and sheep. Thus, the subtherapeutic antibiotic use in ruminant feeding could trigger the formation of antibiotic-resistant mutant colonies, nullifying their subtherapeutic effect [37].

The scientific data available on the nature and composition of ruminal microbial communities in different ruminant species and on their different susceptibility to antibiotics are limited, but the existing literature leads to some important hypotheses concerned the correlation between the development of bacterial resistance with the modification of the bacterial gene profile. In fact, it cannot be excluded that the genes responsible for AMR can be transferred from one bacterium to another in the ruminal site [39,40].

At the ruminal level protozoa are active predators of bacteria that can host genes of resistance to antibiotics, establishing the exchange of genetic material. Sengupta and colleagues identified in Gram-negative bacteria the main reservoir of screened integrons and transposons, even if they do not seem to be responsible for the diffusion of multi-resistance phenotypes of Gram-positive bacteria [41].

The identification of mutant bacterial colonies in the presence of antibiotic treatments underlines their ability to modify and survive [42]. Indeed, bacteria use different mechanisms to resist antibiotics, such as degradation or modification of the chemical structure of the antibiotic, alteration of their bacterial target, and structural modifications aimed at reducing the intracellular concentration of the antibiotic by a decrease in cell wall permeability [43].

Resistance of some mutant isolates seems to be specifically mediated by extracellular polysaccharides (i.e., glycocalyx) that reject antibiotics from the cell membrane [44]. Extracellular polysaccharides play a key role in MON antibiotic resistance of some ruminal bacterial species, such as *Prevotella bryantii* B14 and *Clostridium aminophilum* F [45,46]. In a recent study, the stability, reversibility, and mechanism of MON adaptation of *Enterococcus faecium* (*E. faecium*), *Enterococcus faecalis* (*E. faecalis*) and *Clostridium perfringens* (*C. perfringens*) was analyzed in MON-treated cattle isolates. Microbiological, biochemical analyzes and TEM microscopy demonstrated that MON-exposed isolates grow in the presence of high MON concentrations by increasing the thickening of the cell wall or glycocalyx, suggesting a phenotypically expressed trait [20]. Finally, some other reports indicate that bacterial resistance due to the developed capacity of bacteria to synthetize the outer membrane or to decrease its porosity or by increasing the efflux of the antibiotic from cell [47,48] confers cross-resistance to clinically relevant antibiotics such as the cyclic polypeptide bacitracin or the glycopeptide avoparcin [49].

### 2.2. Cross-Resistance and Co-Selection

It is known that the therapeutic use of antibiotics in animal feed could have an impact on the earlier development of antibiotic resistance in animals and possibly also in humans later. It is important to note that MON is primarily used as a coccidiostat in veterinary medicine and a growth promoter in livestock, particularly in the poultry and cattle industries. It is not used as a first-line antibiotic for treating bacterial infections in humans or animals. Its primary role is controlling parasitic infections caused by coccidia in animals and improving feed efficiency.

Because MON primarily targets certain types of microorganisms, it is less likely to directly contribute to AMR in bacteria that infect humans. However, as previously mentioned, its use in livestock can indirectly influence AMR through complex interactions in the gut microbiota and potential transfer of resistance genes between bacteria leading to cross-resistance and co-selection phenomena (Figure 2). Indeed, although the use of ionophores is limited to veterinary practice, it cannot be assumed that they are safe for human health. Their use may still pose risks due to the possible development of cross-resistance to medically important antibiotics (MIAs) and the co-selection process between ionophores and MIAs, which exist and perhaps have not yet been adequately investigated. This underscores the importance of responsible antibiotic use and ongoing research to understand the broader impacts of antibiotic and ionophore use in agriculture and animal husbandry.

In 1993, Newbold’s work already demonstrated that some species of Gram-negative rumen bacteria developed resistance following treatment with MON and another ionophore, tetronasin [50]. In addition, the increasing resistance to MON led to the development of cross-resistance to a third ionophore lasalocid and to another glycopeptide antibiotic, avoparcin. Similar cross-resistance was registered in rumen samples from MON-fed sheep [50].

An important finding highlights the possibility that vancomycin resistance in Swedish broilers could be maintained using the ionophore narasin. Interestingly, resistance to vancomycin and narasin appears to be based on the localization on the same plasmid of the ABC transporter, which confers putative resistance to narasin, and a *vanA* gene cluster, which confers resistance to vancomycin in enterococci [51].

A recent nucleotide database search conducted by Wong identified 15 strains of *Enterococcus* spp. from different parts of the world and from animal and human samples that present almost identical correspondences to both genes encoding the ABC narasin transporter, and a link to *vanA* was present in some of these strains, supporting this issue [1].

A subsequent surveillance study, which collected data for two years following the abolition of narasin (in 2016 by the Norwegian broiler industry), shows a significant reduction in vancomycin-resistant enterococci and a simultaneous marked reduction in *E. faecium* with reduced susceptibility to narasin in all the isolates from Norwegian broilers fed the food additive narasin from 2006 to 2014 [38]. Importantly, the significant decline in *E. faecium* resistant to these antimicrobial compounds also matched with an increased attention on cleaning and disinfection among broiler farms. Moreover, a controlled in vivo experiment using Ross 308 broilers demonstrated that the percentage of *E. faecium* resistant to narasin was significantly reduced in broilers fed a narasin-free diet compared to a diet supplemented with narasin [38].

Additional antimicrobial susceptibility data demonstrated that the use of MON and tylosin in feedlot cattle and swine results in increased macrolide resistance in fecal foodborne and commensal bacteria of *Enterococcus* spp. [52,53]. Probably, the involvement of the *ermB* gene in enterococci of animal origin is responsible for the development of the common resistance to macrolides by methylating the 23S rRNA and thus making ribosomes tolerant, for example, to erythromycin [54,55].

A recent work published in January 2024 also showed that exposure to different concentrations of MON leads to the selection of resistant mutants of the pathogen of both human and animal interest *Staphylococcus aureus.* A higher growth rate of resistant mutant pathogens was identified both in vitro and in vivo, while genome sequencing and proteomic analysis of resistant mutants showed that the resistance phenotype was associated with de novo de-repression purine synthesis, probably due to mutations in several transcriptional regulators, such as in the *purR* gene [56].

The process of plasmid conjugation represents the main driver through which bacteria can transfer genes that confer resistance to other bacteria, contributing to the dissemination of cross-resistance to critically important antimicrobials. A recent study well documented the effects of residual concentrations of different antimicrobial growth promoters, including MON, in poultry litter on the frequencies of IncFII-FIB plasmid conjugation, which harbors genes conferring resistance to amphenicols, aminoglycosides, β-lactams, macrolides, tetracyclines, trimethoprim and sulphonamides [57] among *Escherichia coli* (*E. coli*) organisms [58]. The presence of residues of MON, lincomycin and virginiamycin increased the frequency of plasmid conjugation among *E. coli* in both types of litter materials (sugarcane bagasse and wood shavings). In addition, the conjugation frequencies were significantly higher in wood shavings compared to sugarcane bagasse in the presence of antimicrobial growth promoters [58].

The increased presence of transconjugant bacteria after treatment with MON, lincomicyn and virginiamycin confirms previous results showing that antimicrobial residues can indeed favor plasmid conjugation [59,60].

These findings support the hypothesis that subinhibitory concentrations of commonly used antimicrobial growth promoters can favor the conjugation of plasmids triggering an acquired genetic framework for AMR important for the survival of bacteria and that the composition of litter significantly impact on conjugation process.

Some of the implications of these findings are related to a possible common mechanism of resistance development among compounds despite the differences in their chemical structure and molecular mechanism of action. Therefore, cross-resistance and co-selection represent real possibilities and the evaluation of their actual impact on AMR in human pathogens deserve to be further investigated together with the environmental implications of the development of cross-resistance and its biochemical and genetic basis [1].

From an ecotoxicological point of view, research on MON fate is limited to only a few European studies, while most of the data come from US, Argentinian and Canadian research.

Among the existing data, several interesting studies have focused on the effects MON may have on the environment due to the application of animal manure as a fertilizer or feed additive or the irrigation of crops with contaminated water. It was demonstrated that MON residues, essentially derived from the parental compound, exert undesired effects on non-target microorganisms leading to selection of resistant bacteria in soil, water bodies or animals [61]. Indeed, MON can persist in cattle manure for up to 10 weeks and has a half-life of 4–13.5 days in soils [18,62]. This environmental behavior results in the accumulation of MON amounts in the order of ng L^−1^, µg/kg^−1^, and mg/kg^−1^, respectively in water, cattle manure, sediment, and soil [63,64,65].

In a recent study, Granados-Chinchilla and colleagues verified whether field-relevant concentrations of MON damage the structure and activity of tropical soil bacteria [49]. The analysis of soil microcosms exposed to 1 or 10 mg kg^−1^ of MON for 11 days showed a subtle concentration-dependent decrease in the number of culturable heterotrophic bacteria. This change was associated with a reduction of CO_2_ efflux and polymers respiration, an increase in respiration rate and amine degradation and a marked shift in non-bacterial fatty acids. The MON concentrations tested in this work, although high, have been occasionally detected in some farm soils [65]. The authors’ hypothesis is that the stress induced by MON exposure rapidly kills some microorganisms while the surviving populations are metabolically activated. Probably, the killed bacteria served as a carbon source increasing the respiration of other microbes. Therefore, even if MON is not applied directly to crops, it reaches soils in its active form through fertilization with manure, which facilitates its antibiotic activity by providing abundant nutrients [66,67]. Furthermore, despite the mentioned half-life of MON of 11 days, its effects have been shown to persist due to the antibiotic activity exerted by its ketone derivatives [68,69].

Interestingly, a study conducted in 2019 performed sorption and mobility analyses to detect the migration capacity of MON, lincomycin and roxarsone in several soil environments and their toxic effects on representative environmental organisms *Scenedesmus obliquus* (algae), *Arabidopsis thaliana* (plant), *Eisenia fetida* (earthworm), *Danio rerio* (zebrafish), *Daphnia magna* (crustacean) and *Coturnix coturnix* (quail) [70]. According to the Kd value (Freundlich model), the absorption rate of laterite to MON was greater than to other drugs tested due to its higher molecular weight and lower water solubility, and its mobility capacity was higher in laterite and black soils. Furthermore, data demonstrate that MON strongly inhibits algal growth and is toxic to daphnia, zebrafish, earthworms, and French giant quail confirming its ability to residual and accumulate in the environment as well as its significantly higher toxicity risk to some species compared to other commonly used antibiotics [71].

## 3. Conclusions and Future Perspectives

The late 20th century witnessed the transfer of resistance from antimicrobials used in animals to essential molecules in human medicine. The use of avoparcin, akin to vancomycin, as a livestock feed additive spurred the rise of vancomycin-resistant bacterial strains. This issue has extended to other antibiotics, presenting a substantial threat to human health. Consequently, avoparcin as a coccidiostatic or growth promoter in animals was banned in the European Union in 1997 [3].

Today, while the scientific literature attests to the benefits of MON in mitigating peripartum diseases in dairy cows without compromising production, and although MON is not employed in human medicine and is less likely to directly contribute to AMR in human pathogens, its administration via controlled-release capsules, releasing the active compound primarily through animal manure for an extended period. This condition raises concerns about the proliferation of antibiotic-resistant genes and AMR development in non-target microorganisms and can influence the resistance gene pool within gut microbiota across individual animals. Ongoing research seeks to elucidate these intricate mechanisms and outcomes, which may differ according to factors such as animal species, dosage, and other variables. Moreover, the long-term consequences of these changes are of growing interest, encompassing animal health, productivity, environmental, and public health concerns.

Thus, an in-depth evaluation is warranted to assess whether the economic advantages justify the use of MON, its local and global environmental impact, and the risk of AMR.

The use of MON and other ionophores in livestock requires careful management and oversight to mitigate the potential repercussions on animal health, welfare, and the environment, all while considering broader implications for AMR and gut microbiota ecology. It is paramount to acknowledge that antibiotics cannot substitute sound animal management practices or tailored diets aligned with livestock production cycles. The preservation of animal welfare, the environment, and human health should remain the primary and overriding objectives. Therefore, ionophores, including MON, should be subject to rigorous analytical monitoring within agronomic and environmental systems as part of One Health initiatives. This is especially crucial in countries with intensive cattle and poultry production systems to prevent the exacerbation of bacterial cross-resistance, posing a deeper public health challenge.

In conclusion, due to MON’s extensive biological activity and the potential for AMR spread to crucial human-use molecules, its pharmaceutical forms and uses must be closely regulated to align with the latest World Health Organization (WHO) guidelines [72].

Overall, the use of antimicrobials in food-producing animals should be reduced, with a complete restriction on their use for growth promotion and disease prevention, aligning with the European directive 2019/6, which particularly highlights the responsible use of antimicrobial [73].

A One Health approach is endorsed to combat AMR, necessitating collaborative efforts from veterinarians, farmers, regulatory agencies, and all stakeholders to implement these guidelines. Therefore, the use of antimicrobial-potential molecules in animals should be judiciously monitored and authorized only when necessary.

## Figures and Tables

**Figure 1 antibiotics-13-00129-f001:**
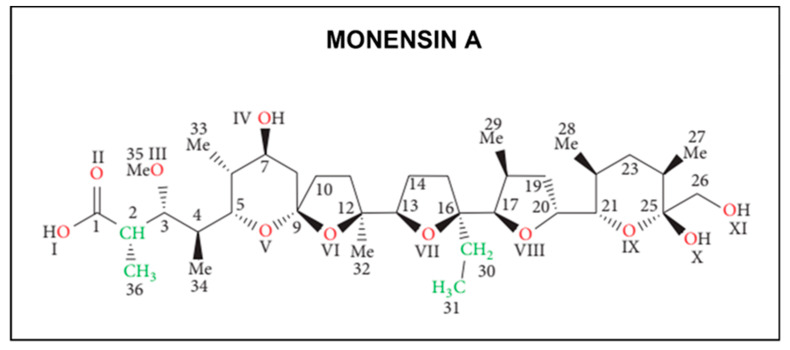
The chemical structure of Monensin A.

**Figure 2 antibiotics-13-00129-f002:**
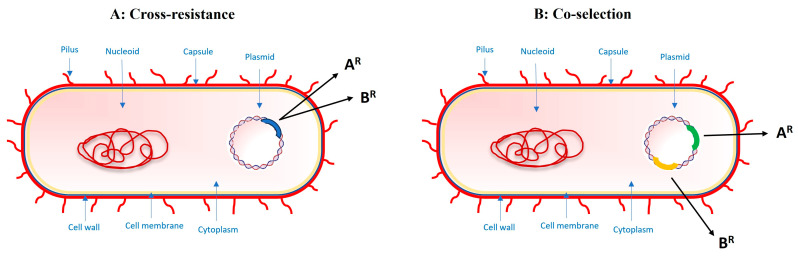
(**A**) A single mutation or gene may confer cross-resistance to two or more drugs. (**B**) Alternatively, co-selection can occur if separate resistance mutations or genes for several drugs are genetically linked, and thus selection with one drug will co-select for resistance to a second drug. A^R^ = resistance gene A; B^R^ = resistance gene B. Figure modified from (Wong A. mSphere 2019 [1]).

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
