# Peer review of "Is the Use of Monensin Another Trojan Horse for the Spread of Antimicrobial Resistance?"

_antibiotics, 2024, doi:10.3390/antibiotics13020129_

Round 1
Reviewer 1 Report
Comments and Suggestions for Authors
Summary: In the article by Carresi et al., the authors examine the potential implications and risks associated with the use of monensin as a coccidiostat in industrial poultry farming and as a preventive measure against ketosis in lactating cows. While the use of monensin in recommended doses is relatively safe, its overuse can trigger the risk of antimicrobial resistance or cross-resistance.
Review : Carresi et al.'s article significantly contributes to the understanding of the nuanced relationship between monensin application, recommended doses, and associated risks. The comprehensive exploration of its dual role as a coccidiostat and a preventive measure in different livestock contexts provides valuable insights for practitioners, veterinarians, and policymakers alike. The emphasis on responsible use emerges as a crucial takeaway, urging a careful balance to harness the benefits of monensin while mitigating the potential risks that could compromise animal health and contribute to broader concerns such as antimicrobial resistance.
Author Response
Point-by-point Authors’ Rebuttal to Reviewers’ Comments Reviewer #1
Summary: In the article by Carresi et al., the authors examine the potential implications and risks associated with the use of monensin as a coccidiostat in industrial poultry farming and as a preventive measure against ketosis in lactating cows. While the use of monensin in recommended doses is relatively safe, its overuse can trigger the risk of antimicrobial resistance
or cross-resistance.
Review: Carresi et al.'s article significantly contributes to the understanding of the nuanced relationship between monensin application, recommended doses, and associated risks. The comprehensive exploration of its dual role as a coccidiostat and a preventive measure in different livestock contexts provides valuable insights for practitioners, veterinarians, and policymakers alike. The emphasis on responsible use emerges as a crucial takeaway, urging a careful balance to harness the benefits of monensin while mitigating the potential risks that could compromise animal health and contribute to broader concerns such as antimicrobial resistance.
We sincerely thank the Reviewer for her/his kind comment which we fully share regarding the contents and purposes that our article aims to disclose.

Reviewer 2 Report
Comments and Suggestions for Authors
In this manuscript, Carresi et al. provide a brief overview of how the use of agricultural and environmental ionophore monensin (MON) may contribute to the spread and persistence of antimicrobial resistance (AMR) and have an impact on human health. The authors briefly reviewed the chemical structure and the antimicrobial activity of MON, the concept of the antimicrobial resistance, and the potential mechanisms by which the use of MON could contribute to the development of microbial AMR genes or phenotypes. Overall, the manuscript is well-written and offers a compelling perspective to a controversial and underaddressed topic. Below I only have some minor suggestions for the authors to consider:
- In lines 198-199, the authors need to clarify what they mean by this sentence: “The identification of surviving mutant bacterial colonies in the presence of antibiotics suggests an inhibition of their growth but not their death [41]”. Of course the antibiotics does not inhibit the death of the bacteria. Maybe the authors simply want to say that a subpopulation can survive in the presence of antibiotics?
- In several places throughout the manuscript, the authors refer to some previous review articles to support statements, such as referencing [43] in line 216 and referencing [57] in line 307. My suggestion is the authors should always refer to the original research articles that demonstrated certain findings.
- For section 2.2, it would be beneficial to include a cartoon figure illustrating what cross-resistance and co-selection look like, since they are really the main takeaways of the current review. I would strongly suggest adding something similar to Fig. 1 of [Wong, mSphere 2019].
- Minor suggestion: I don’t think the authors very rigorously distinguish between antimicrobial resistance, tolerance, and persistence (see [Brauner et al., Nat Rev Microbiol 2016]). If the authors indeed refer to all these scenarios as “”antimicrobial resistance”, it might be helpful to clarify in the very beginning that “resistance” is loosely defined in the current manuscript.
- Minor suggestion: Since this is a new review, I would encourage the authors to reference more articles published in the last few years to reflect the current status of the research area.
The authors need to check for typos and grammar issues throughout the manuscript. For example, in line 272, the sentence “…represents the main driven through which…”, seems to contain a typo – “driven” should be changed to “driver”.
Author Response
Point-by-point Authors’ Rebuttal to Reviewers’ Comments
We thank Reviewer for her/his comments. We have carefully addressed each of her/his comments in the revision of our paper and we believe to have satisfactorily followed all her/his suggestions as below detailed.
Reviewer #2
In this manuscript, Carresi et al. provide a brief overview of how the use of agricultural and environmental ionophore monensin (MON) may contribute to the spread and persistence of antimicrobial resistance (AMR) and have an impact on human health. The authors briefly reviewed the chemical structure and the antimicrobial activity of MON, the concept of the antimicrobial resistance, and the potential mechanisms by which the use of MON could contribute to the development of microbial AMR genes or phenotypes. Overall, the manuscript
is well-written and offers a compelling perspective to a controversial and underaddressed topic. Below I only have some minor suggestions for the authors to consider:
1. In lines 198-199, the authors need to clarify what they mean by this sentence: “The identification of surviving mutant bacterial colonies in the presence of antibiotics suggests an inhibition of their growth but not their death [41]”. Of course the antibiotics does not inhibit the death of the bacteria. Maybe the authors simply want to say that a subpopulation can survive in the presence of antibiotics?
The sentence in lines 198-199 (now 205-207) has been corrected as suggested.
2. In several places throughout the manuscript, the authors refer to some previous review articles to support statements, such as referencing [43] in line 216 and referencing [57] in line 307. My suggestion is the authors should always refer to the original research articles that demonstrated certain findings.
References have been modified as suggested.
3. For section 2.2, it would be beneficial to include a cartoon figure illustrating what crossresistance and co-selection look like, since they are really the main takeaways of the current review. I would strongly suggest adding something similar to Fig. 1 of [Wong, mSphere 2019].
The figure in section 2.2 has been added as suggested.
4. Minor suggestion: I don’t think the authors very rigorously distinguish between antimicrobial resistance, tolerance, and persistence (see [Brauner et al., Nat Rev Microbiol 2016]). If the authors indeed refer to all these scenarios as “antimicrobial resistance”, it might be helpful to clarify in the very beginning that “resistance” is loosely defined in the current manuscript.
The meaning of “antimicrobial resistance” has been better clarified in the Introduction section as suggested.
5. Minor suggestion: Since this is a new review, I would encourage the authors to reference more articles published in the last few years to reflect the current status of the research area.
More recent references have been added as suggested.
6. The authors need to check for typos and grammar issues throughout the manuscript. For example, in line 272, the sentence “…represents the main driven through which…”, seems to contain a typo – “driven” should be changed to “driver”.
A typos and grammar check has been performed as suggested.

Round 2
Reviewer 2 Report
Comments and Suggestions for Authors
I'm satisfied by the changes that the authors made, which addressed my previous comments.
Comments on the Quality of English LanguageIt's fine at this stage.